# Peering into the Darkness: DNA Barcoding Reveals Surprisingly High Diversity of Unknown Species of Diptera (Insecta) in Germany

**DOI:** 10.3390/insects13010082

**Published:** 2022-01-12

**Authors:** Caroline Chimeno, Axel Hausmann, Stefan Schmidt, Michael J. Raupach, Dieter Doczkal, Viktor Baranov, Jeremy Hübner, Amelie Höcherl, Rosa Albrecht, Mathias Jaschhof, Gerhard Haszprunar, Paul D. N. Hebert

**Affiliations:** 1SNSB-Zoologische Staatssammlung München, Münchhausenstr. 21, 81247 München, Germany; hausmann.a@snsb.de (A.H.); schmidt.s@snsb.de (S.S.); raupach@snsb.de (M.J.R.); doczkal@snsb.de (D.D.); huebner@snsb.de (J.H.); hoecherl@snsb.de (A.H.); albrecht@snsb.de (R.A.); haszprunar@snsb.de (G.H.); 2Department Biology II, Ludwig-Maximilians-University of Munich (LMU), Großhaderner Str. 2, Martinsried, 82152 Planegg, Germany; baranowiktor@gmail.com; 3Station Linné, Ölands Skogsby 161, 38693 Färjestaden, Sweden; mjaschhof@yahoo.de; 4Centre for Biodiversity Genomics, University of Guelph, Guelph, ON N1G 2W1, Canada; phebert@uoguelph.ca

**Keywords:** Diptera, insects, dark taxa, taxonomic impediment, species estimates, DNA barcoding, biodiversity, German insect fauna

## Abstract

**Simple Summary:**

Roughly two-thirds of the insect species described from Germany belong to the orders Diptera (flies) or Hymenoptera (wasps, bees, ants and sawflies). However, both orders contain several species-rich families that have received little taxonomic attention until now. This study takes the first step in assessing these “dark taxa” families and provides species estimates for four challenging groups of Diptera (Cecidomyiidae, Chironomidae, Phoridae and Sciaridae). The estimates given in this paper are based on the sequencing results of over 48,000 fly specimens that have been collected in southern Germany via Malaise traps that were operated for one season each. We evaluated the fraction of species in our samples belonging to well-known fly families in order to estimate the species richness of the challenging “dark taxa” (DT families hereafter). Our results suggest a surprisingly high proportion of undetected biodiversity in a supposedly well-investigated country: at least 1800–2200 species await discovery and description in Germany in these four families.

**Abstract:**

Determining the size of the German insect fauna requires better knowledge of several megadiverse families of Diptera and Hymenoptera that are taxonomically challenging. This study takes the first step in assessing these “dark taxa” families and provides species estimates for four challenging groups of Diptera (Cecidomyiidae, Chironomidae, Phoridae, and Sciaridae). These estimates are based on more than 48,000 DNA barcodes (COI) from Diptera collected by Malaise traps that were deployed in southern Germany. We assessed the fraction of German species belonging to 11 fly families with well-studied taxonomy in these samples. The resultant ratios were then used to estimate the species richness of the four “dark taxa” families (DT families hereafter). Our results suggest a surprisingly high proportion of undetected biodiversity in a supposedly well-investigated country: at least 1800–2200 species await discovery in Germany in these four families. As this estimate is based on collections from one region of Germany, the species count will likely increase with expanded geographic sampling.

## 1. Introduction

Although the Central European insect fauna is considered to be well studied, gaps in knowledge of its taxonomy and biodiversity remain [1]. About 33,300 species of insects are documented from Germany, of which roughly two-thirds of these taxa belong to one of the two orders: Diptera (flies) and Hymenoptera (wasps, bees, ants, and sawflies) [1,2,3,4,5,6,7,8]. However, both orders contain several species-rich families which have received less attention than others in Germany’s long history of taxonomic research [1]. This reflects the confluence of several factors, such as extreme species richness combined with a high rate of cryptic diversity and, most importantly, the limited taxonomic attention directed to small specimens (<2 mm) whose morphological characteristics are difficult to evaluate. Successful identification of species in these groups using morphology is time-consuming and requires taxonomic expertise, the availability of which is decreasing [9,10,11,12,13,14]. This imbalance of few researchers but high species numbers still awaiting documentation is commonly referred to as the taxonomic impediment [9,15,16]. Against the backdrop of a worldwide decline in insect abundance, the taxonomic impediment is an alarming constraint to biodiversity surveys [17,18,19,20,21]. One such constraint is noticeable in the framework of DNA barcoding applications, where species proxies (Barcode Index Numbers, BINs) often lack a linkage to a known species [22]. Page [22] coined the term “dark taxa” for these nameless BINs, and in 2020, Hausmann et al. [1] used it to address species-rich, taxonomically challenging groups of insect families whose diversity remains mostly undescribed. These include certain families of non-brachyceran Diptera (mosquitoes, gnats, midges), some families of Brachycera (flies), and nearly all families of parasitoid Hymenoptera (wasps) which often make up the majority of the insect biodiversity present in environmental and bulk samples [23]. With the shortage of taxonomic specialists, the functional role of “dark taxa” in ecosystems is far too understudied, meaning that they cannot be included in biomonitoring or conservation surveys.

The most recent project in the German Barcode of Life initiative, GBOL III: Dark Taxa, was launched in mid-2020 to tackle these challenging groups. Its two main goals are: (1) to study various DT families using an integrative taxonomic approach which combines morphological and sequence data [1,24], and (2) to expand the DNA barcode reference library established by three earlier initiatives (Barcoding Fauna Bavarica, GBOL I, GBOL II) [24,25,26]. Work conducted by GBOL II generated a reference library for the order Diptera based on 50,963 COI sequences, data that provided barcodes for 5200 BINs [13]. A recent commentary on this study presented a classical dipterist’s perspective on the situation for the better-known families of Diptera [27]. It explored ways to extend the involvement of expert taxonomists in assigning Linnean names to BINs. However, the challenge in implementing similar work on DT families was not addressed, highlighting the need to seek new approaches so these taxa can finally become more accessible to research.

This study begins this effort by considering the German fauna of four DT families of Diptera which lack estimates of their species numbers: Cecidomyiidae (gall midges), Chironomidae (non-biting midges), Phoridae (scuttle flies), and Sciaridae (dark-winged fungus gnats) (Figure 1). To address this goal, we examine the diversity of these DT families in our Malaise trap collections. We employ BIN data resulting from the sequence analysis of samples from southern Germany and use these results to estimate the extent of undocumented biodiversity in these families in Bavaria and Germany. An important backbone to our calculations is species numbers inferred from essential contributions of Germany’s over 200-year-long history of taxonomy [5,6,7,8,28,29,30,31,32,33,34,35,36,37,38].

## 2. Materials and Methods

### 2.1. Malaise Tap Sites

In 2012, the Global Malaise Trap Program was launched by the Centre for Biodiversity Genomics (CBG) at the University of Guelph to provide a global overview of arthropod diversity [39]. As part of this project, 14 Malaise traps were deployed at various sites in Germany (Figure 2 and Table 1). In 2012, one trap was operated from May to September in the Bavarian Forest National Park (BFNP), a conifer-dominated montane forest. In 2014, 12 Malaise traps were placed along an altitudinal transect (1036–2160 m) in the Allgäu Alps, ranging from the Oytal to the Schochen and Nebelhorn Mountains. Traps in lower altitudes (Oytal) were deployed in May, whereas those in higher altitudes (Schochen and Koblat) were deployed in June. All traps in the Allgäu Alps were operated until October. Finally, in 2017, one trap was deployed at the Bavarian State Collection of Zoology (ZSM) in Munich, which is situated in a residential neighborhood rich in backyard gardens. This trap was operated from April to December. Altogether, the sampled sites represent a heterogeneous array of habitats typical of southern Germany. The specifics of trap deployment (habitat type, site, orientation, height) strongly influence its catch [40]. Collection dates varied among sites but are detailed in Table A1. Denatured ethanol (80%) was used to preserve specimens.

### 2.2. Processing of Specimens

Samples from two sites (BFNP, ZSM) were sent directly to the CBG for analysis. Due to funding constraints, roughly every second weekly sample from the BFNP and every fourth weekly sample from the ZSM were selected for DNA barcode analysis. Based on the number of specimens in the samples that were processed, the full year of collecting at these sites yielded about 52,000 and 130,000 specimens, respectively. Using morphology, specimens from these locales were sorted to an order prior to sequence analysis and to a family after analysis. In total, tissue samples or whole individuals of 62,073 specimens (29,481 from BFNP; 32,592 from ZSM) were transferred to 96-well microplates for DNA extraction. Samples from the Allgäu Alps were sorted by a dipterist at the ZSM before being dispatched in 96-well microplates to the CBG for sequence analysis. Rough estimates suggest the Allgäu samples included well over a million specimens, but funding was only available to process about 2% of them (20,250 specimens).

At the CBG, specimens were processed using standard protocols for DNA extraction, PCR amplification of the barcode region of COI, and sequencing. Specimens from the BFNP and the Allgäu Alps were Sanger sequenced on an ABI 3730XL [41], while specimens from the ZSM were sequenced on Sequel [42].

### 2.3. Data Analysis

All specimen metadata and sequence data were uploaded to the Barcode of Life Data System (BOLD), an online workbench and database [32]. These data are publicly available in three datasets: DS-BFNP, DS-ZSMTRAP and DS-ALGALPS. Each sequence ≥ 300 base pairs (bp) was automatically assigned to a Barcode Index Number (BIN) already in BOLD if sequence similarity based on the (RESL-) BIN algorithm was fulfilled [43]. Sequences ≥ 500 bp which did not find a match served as founders of new BINs. All data were downloaded on 8 February 2021 for further analysis. Therefore, the present results correspond to BINs assigned at that time (BIN assignments can change as new sequences are added to BOLD).

Employing BINs as a proxy for species, we employed Chao1 [44] to estimate species counts for the dipteran families selected for analysis. We then calculated the ratio between the observed number of BINs in our samples to the estimate of species richness generated by Chao1 to ascertain the proportion of species at the sampling sites that have not been captured by our Malaise traps and that await analysis. We also generated continuous diversity profiles that illustrated variation in three standard metrics of biodiversity, which are quantified by Hill numbers (q): species richness (q = 0), Shannon diversity (q = 1), and Simpson diversity (q = 2) [34]. Hill numbers are a mathematically consolidated group of diversity indices which include relative species abundances in order to quantify biodiversity [45]. All calculations were performed in R version 3.3.6 with the Chao1 estimates calculated using the *SpadeR* package [46].

### 2.4. Extrapolating Species Numbers

We selected, more or less randomly, 11 dipteran families whose taxonomy and fauna have been intensively studied to date in order to assess the fractions of the Bavarian and German faunas represented in our samples (Table 2). By comparing the known species counts for these 11 families with the species recovered from our Malaise traps, we could estimate the percentage of these taxa that were recovered, providing a basis for estimating the completeness of our sampling. These values could then be used to estimate species diversity for our four DT families: Cecidomyiidae—gall midges; Chironomidae—non-biting midges; Phoridae—scuttle flies, and Sciaridae—dark-winged fungus gnats.

Species numbers for Germany and for Bavaria were obtained from extensive literature (Table 2). For each family where a species count for Bavaria was unavailable, we adopted a count equal to 0.80 of the species number for Germany. This value was conservative because where species lists were available for both Bavaria and Germany, the ratio often exceeded 0.80 (Table 2). Moreover, this proportion corresponds to past evidence that Bavaria hosts 80–85% of the German fauna in well-studied invertebrate groups, both terrestrial and limnic [2,47].

We estimated species numbers for the DT families through the following steps:We calculated a Recovery Ratio by dividing the number of BINs detected through sequencing by the species count for each of the 15 families and for all Diptera (BIN/species ratio). This approach generated a ratio for each well-known family, for each DT family, and for all Diptera.We estimated the maximum number of species for each “dark taxon” for both Germany and Bavaria by dividing its BIN count by the average BIN/species ratio of all 11 well-known families.We estimated the minimum species number for each “dark taxon” by dividing all Diptera BINs by all Diptera species (i.e., 9544). Because this calculation includes numerous families with cryptic diversity, the resultant values underestimate the diversity of the DT families.

In the same fashion, we extrapolated species numbers employing the Chao1 values for the four DT families.

## 3. Results

### 3.1. Sequencing Results

COI sequences were recovered from 85.4% of the insects (70,293/82,323) that were analyzed (Table 3) and success was even higher for Diptera (91%). Diptera comprised nearly two thirds of the specimens that were analyzed and more than half of the resultant BINs. When results for Diptera from the three collection sites were pooled, the resulting 48,230 COI sequences were assigned to 4863 BINs and included species from 85 families. Across all sites, roughly 20% of the BINs were new to BOLD and almost 70% of them were Diptera with representatives from 56 families. Almost half of all dipteran BINs (2146; 44.1%) and 55% of the new dipteran BINs belonged to the four DT families.

### 3.2. Estimation of Taxon Diversity Using BIN/Species Ratios

The 11 well-known families of Diptera displayed BIN/species ratios that ranged from 0.19–0.60 (ø 0.33 ± 0.9) for Bavaria and from 0.15–0.48 (ø 0.27 ± 0. 7) for Germany (Table 4, Figure A1a). Dividing all Diptera BINs by all known Diptera species produced a ratio of 0.64 for Bavaria and 0.51 for Germany. While one DT family (Chironomidae) possessed a ratio (0.38, Germany) that overlapped the upper end of the values for the 11 well-known families, the other three had far higher ratios. In fact, the BIN count for Phoridae and Sciaridae nearly matched the known species count for Germany, while the count for Cecidomyiidae exceeded it.

### 3.3. Estimation of Taxon Diversity Using Chao1/Species Ratios

Chao1 estimates of species richness were obtained for the 15 families of Diptera (Table 5). BIN/Chao1 ratios averaged 0.76 for the 11 well-known families. The diversity profiles for 10 of these families showed overlap between the species richness in our samples and that estimated to occur at the sites sampled by our Malaise traps (Hill number q = 0, Figure 3). Muscidae was the sole exception as its predicted diversity was considerably higher than currently recognized. Chao1/species ratios ranged from 0.21–0.82 (0.46 ± 0.2) for Bavaria and from 0.16–0.66 (0.37 ± 0.2) for Germany (Table 5).

The BIN/Chao1 ratios for the DT families were similar to those for the well-known families, ranging from 0.60–0.81 (ø 0.69 ± 0.8). The diversity profiles for all four families (Figure 4) showed no overlap between observed and estimated species richness (i.e., Hill number q = 0). Chao1/species ratios indicated coverages of 0.83–5.91 for Bavaria and 0.61–2.25 for Germany (Table 5). Excluding Chironomidae, all DT families possessed ratios well above 1. Considering all Diptera, our samples recovered about 70% of the species estimated to occur at the study sites, meaning that as many as 6927 BINs of Diptera could have been collected during sampling. Chao1/species ratios were 0.91 for Bavaria and 0.73 for Germany.

### 3.4. Extrapolating Species Numbers

We employed the two ratios to estimate the number of species in the DT families. First, we used BIN/species ratios to extrapolate species numbers based on the number of observed BINs. Second, we used the Chao1/species ratios to estimate species numbers based on the estimated BIN diversity. The first approach generates more conservative values than the second. We divided the number of observed BINs by the (BIN or Chao1)/species ratio for all Diptera to calculate minimum species numbers. To obtain an upper limit, we divided the number of observed BINs for each family by the average (BIN or Chao1)/species ratio for all well-known families. The following calculation is presented below (e.g., Sciaridae).

As 339 Sciaridae BINs were recovered, the minimum species estimate for Bavaria was 530 (339/0.64), while the upper estimate was 1027 (339/0.33). Similarly, the number of species in Germany could be estimated as ranging from 665 (339/0.51) to 1255 (339/0.27) species. By making similar calculations for each DT family, an overall estimate for total species numbers in Bavaria and Germany was obtained (Table 6). The number of species that await discovery in each region can then be obtained by subtracting the number of known species from these estimates.

In total, we recovered 2146 BINs for the DT families which is 22% of the total count of dipteran species known from Germany. Our conservative estimate suggested that just the DT families comprise about 3300–6500 species in Bavaria versus 4200–7900 in Germany. Based on the current species count for Diptera in Bavaria (7635) and Germany (9544), and our estimate of new record, this implies an increase of 25–66% and by 19–59% respectively.

By comparison, the Chao1 analysis suggested that 3316 BINs of the DT families occurred at our sampling sites, a 54% increase from current estimates. Based on this approach, there about 2200–5800 species in Bavaria and 2200–6600 in Germany that may still await documentation. Hence, this approach raises the species count for Diptera by 29–75% for Bavaria and by 22–69% for Germany.

## 4. Discussion

Although members of the order Diptera comprise almost a third of Germany’s insect fauna, the true diversity of the four highly diverse families [1] examined in this study is likely much higher than previously assumed [13,38]. By assessing the number of BINs sequenced from our collections and extrapolating species numbers, we obtained an initial estimate of their species numbers. Our results suggest that at least 1900–2200 dipteran species await discovery in Bavaria versus 1800–2200 in Germany. Although our species estimates were only based on sequencing Bavarian specimens, they are likely a good approximation of diversity in Germany as 80–85% of the invertebrate species found in Germany occur in Bavaria [2,36]. While Bavaria does have some habitats (e.g., alpine) that are not found in other regions of Germany, other habitats (e.g., coastal marshes) are absent [2], meaning that species specialized in the latter habitats will not occur in the state.

### 4.1. DNA Barcoding: Using BIN Numbers as Proxies for Species Numbers

Prior studies [49] have demonstrated that DNA barcoding is not only effective for specimen identification, but is also valuable for estimating species numbers [50,51,52,53]. Although there is strong correspondence between BIN counts and species numbers [49,54], several factors can lead to differences [54]. For example, COI numts can lead to the overestimation of species numbers if they are preferentially amplified in some specimens [55,56,57,58]. Conversely, the introgression of mitochondrial DNA (mtDNA), incomplete lineage sorting, and recent speciation can lead to underestimation of species numbers [59,60,61]. Other factors that challenge COI-based species identifications include heteroplasmy [62] and the homogenization of mtDNA haplotypes due to the maternally inherited endosymbiont *Wolbachia* [63,64]. These underlying molecular factors can lead the BIN algorithm on BOLD to assign members of a single species to several BINs or to assign several species to a single BIN. In groups with well-developed taxonomic systems, the BIN algorithm typically underestimates the true species count by about 10% as it was designed to deliver a conservative value for species diversity [65]. In addition to this internal constraint, two operational factors may have led our study to substantially underestimate actual species numbers:Limited geographic sampling as our data originates from few sites in Bavaria only, covering a tiny fraction of habitat types otherwise present.Limited funding constrained analysis to just 5% of the 1.2 million specimens that were collected.

### 4.2. BIN & Chao1/Species Ratios: Well-Known Families versus DT Families

We assessed the completeness of the species coverage provided by our Malaise trap samples in two ways. First, we calculated the ratio of the BINs recovered for each family and its known species count for Bavaria and Germany. We then made the same calculation employing Chao1 estimates, which, in contrast to the first approach, includes species that were present at our sampling sites but not caught nor sequenced. Thus, it is important to note that our first approach generates more conservative values than the second. By calculating the BIN/Chao1 ratios for each taxon, we were able to make the proportion of diversity that was not captured tangible.

Overall, the resulting (BIN or Chao1)/species ratios were much higher for the DT families than for the well-known ones (Table 4 and Table 5). Average ratios among the well-known families were well under 1 (ranging from 0.33–0.46 for Bavaria and 0.27–0.37 for Germany), indicating that our collections only included a fraction of the known diversity from Bavaria and Germany. This was expected because we only sampled few sites and only processed a fraction of our dipteran specimens. The much higher ratios for the DT families (average ranging from 1.67–2.55 for Bavaria and 0.91–1.34 for Germany) strongly suggest the presence of undescribed, unknown species. The Cecidomyiidae were the most dramatic case as we detected 1163 BINs, a value 35% higher than the species count for this family in Germany [8]. In fact, a quarter of all Diptera BINs belonged to this family, reinforcing conclusions from earlier studies indicating that this is the most diverse family of flies [13,49]. For example, extensive sampling at sites across Canada [49] revealed more than 10,000 BINs, a result which suggested that the Cecidomyiidae may include two million species worldwide. The Bavarian fauna has received little taxonomic attention as only 328 species are recorded versus a likely count of 687 species based on the presumption that 80% of the German fauna occurs there. By contrast, our analysis of 7148 specimens revealed 1163 BINs, a count for Bavaria which is threefold higher than the number of recorded species. Chironomidae was an exception among our DT families, as we obtained ratios that were consistent with those of the well-known families (Table 5). Although Chironomidae is a dark taxon, extensive research concerning the systematics, taxonomy, and nomenclature of European and Neotropical species has and is being conducted at the Bavarian State Collection of Zoology (ZSM) by the late Ernst Fittkau (former director of the ZSM) and his students including Martin Spies, the current editor of the Chironomid Home Page [66]. We therefore expect that the chironomid fauna of Bavaria and Germany is well documented and that, in contrast to the other DT families, a much lower amplitude of new species will be discovered in the following years of GBOL III. Among the well-known families, the Muscidae displayed the highest BIN/species indicating that the current species count considerably underrepresents its actual diversity. As a result, the Muscidae should also be recognized as a DT family.

### 4.3. Discrepancies in Taxa Coverage in Our Malaise Traps

Our estimated species counts for the DT families are based on the presumption that recovery success for the 11 families with strong taxonomy is a useful predictor of recovery success for the DT families. Our results did reveal threefold differences in recovery success among the well-known families, being lowest for Asilidae and Tabanidae and highest for the Muscidae. In our study, we used Malaise traps as a source of insect material, because they enable sampling of high numbers of flying insects, especially Diptera [67,68,69]. However, a bias favoring the sampling of some taxa over others is always present, meaning that the community captured with such traps does not depict the true insect community of a sampled site [67]. Furthermore, the setup of a Malaise trap in terms of site choice, orientation, and above-ground-level is another source of bias, and these factors strongly influence sampling results [40]. To incorporate such variations, we used different approaches for extrapolating species numbers including Chao1 estimate calculations, which consider the unsampled taxa present at the sampling sites. The resulting Chao1 values indicated that we only recovered about 70% of the dipteran species present at the sites. In this manner, we obtained BIN estimates for each family that consider recovery success and unsampled taxa. Our results indicate that more than 3316 more BINs await detection, a total that would raise the number of Dipteran species in Germany by a third.

## 5. Conclusions

In this study, we aimed at estimating the number of species in the Bavarian and German faunas for four families of Diptera that are prime examples of “dark taxa”. Our estimates were inferred from the analysis of sequence data, reproducible genetic patterns, rather than on speculations. The confidence intervals on these estimates are broad (Table 5), reflecting the various factors that influence any effort to gauge species diversity. Despite our limited geographic sampling effort, our results strongly suggest that a surprisingly high proportion of Germany’s biodiversity is yet to be discovered.

## Figures and Tables

**Figure 1 insects-13-00082-f001:**
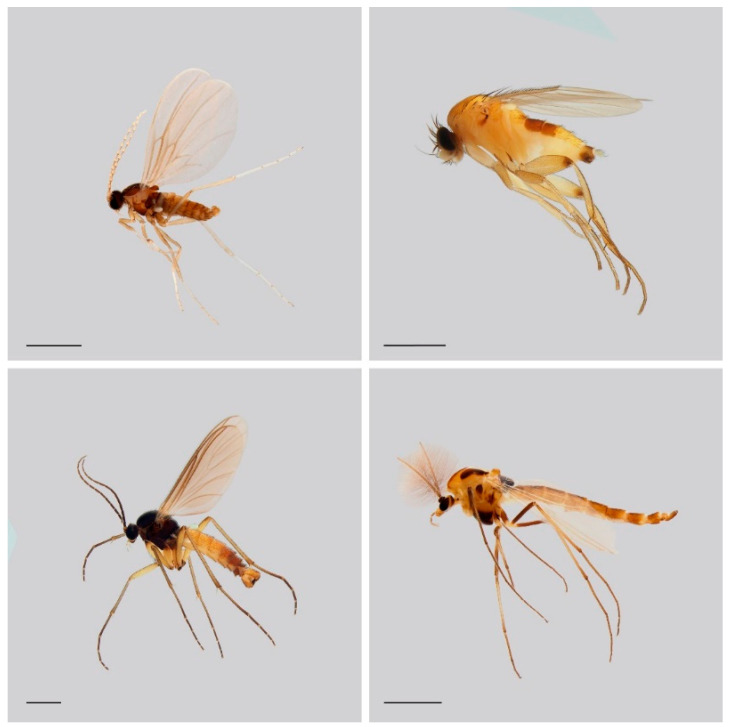
Selected representatives of the DT families analyzed in our study: Cecidomyiidae (**top left**); Phoridae (**top right**); Sciaridae (**bottom left**) and Chironomidae (**bottom right**). Scale bars represent 1 mm.

**Figure 2 insects-13-00082-f002:**
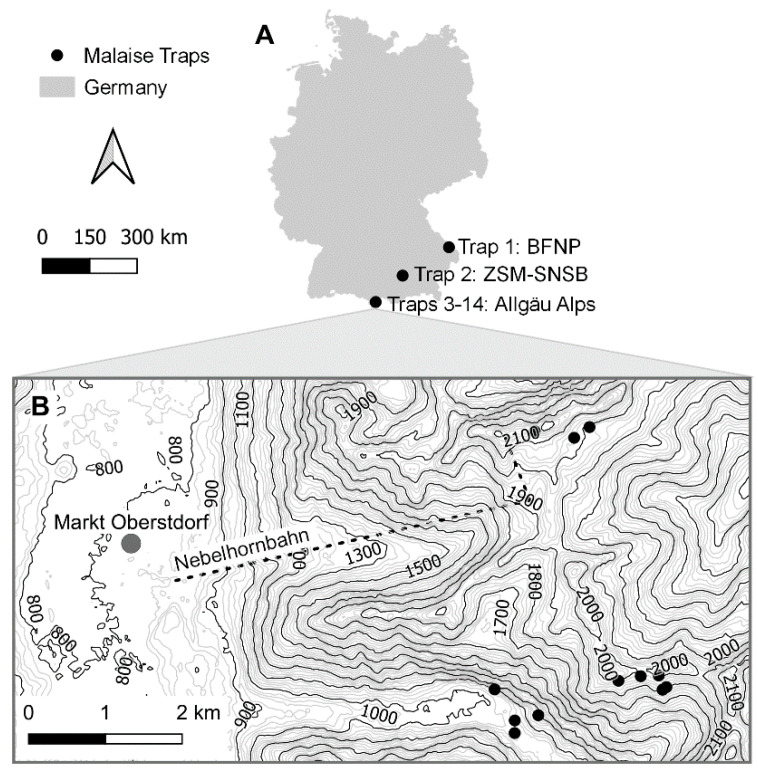
Malaise trap sites. Locations where the 14 Malaise traps were deployed in 2012, 2014, and 2017 ((**A**,**B**) shows enlarged map of Allgäu Alps) as Germany’s contribution to the Global Malaise Trap Program.

**Figure 3 insects-13-00082-f003:**
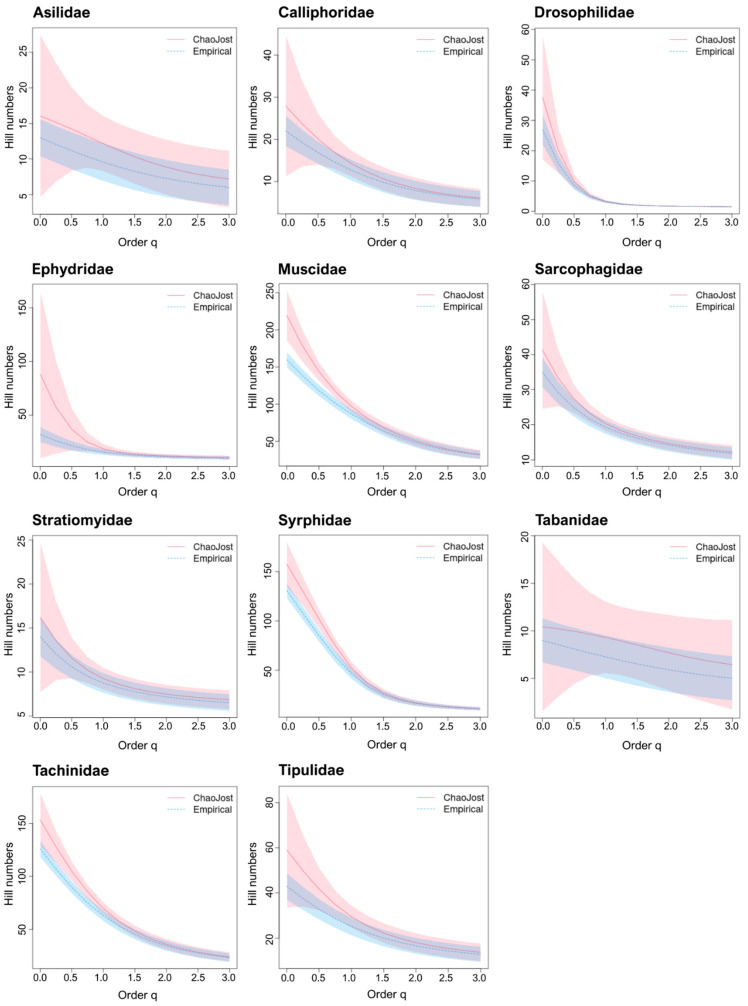
Diversity profiles for 11 well-known taxa. The empirical (BIN counts; dotted blue) and estimated (Chao1; red) diversity profiles for communities where Malaise traps were deployed, as quantified by Hill numbers for each of the 11 well-known families for values of the diversity order (q) from 0–3 with 95% confidence intervals (shaded areas based on bootstrap analysis of 100 permutations). Species richness is depicted by q = 0; Shannon diversity by q = 1; and Simpson diversity by q = 2.

**Figure 4 insects-13-00082-f004:**
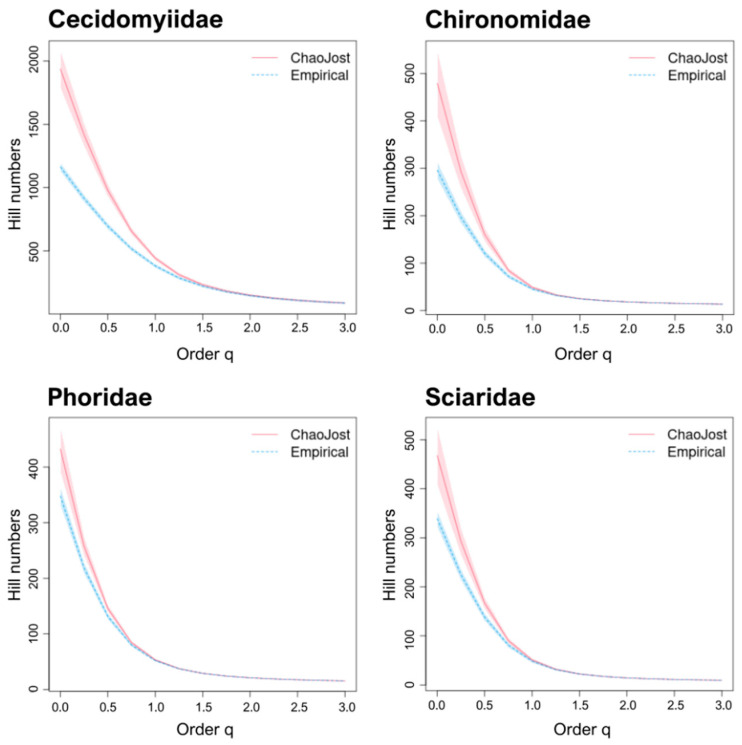
Diversity profiles for the four DT families. The empirical (BIN counts; dotted blue) and estimated (Chao1; red) diversity profiles for communities where Malaise traps were deployed, as quantified by Hill numbers for each of the four “dark taxa” families for values of the diversity order (q) from 0–3 with 95% confidence intervals (shaded areas based on bootstrap analysis of 100 permutations). Species richness is depicted by q = 0; Shannon diversity by q = 1; and Simpson diversity by q = 2.

**Table 1 insects-13-00082-t001:** Malaise trap information. Trap site, exact location, elevation, and habitat type.

Site	Trap	Coordinates	Elevation	Habitat
BFNP	Trap 1	48.9509° N 13.422° E	842 m	Natural forest
ZSM	Trap 2	48.1648° N 11.4849° E	519 m	Urban, pre-alpine meadow
Allgäu Alps: Oytal	Trap 3	47.39205° N 10.34093° E	1122 m	Lake rock face
Allgäu Alps: Oytal	Trap 4	47.38903° N 10.34846° E	1200 m	Cone of scree
Allgäu Alps: Oytal	Trap 5	47.38842° N 10.34440° E	1056 m	Rough pasture
Allgäu Alps: Oytal	Trap 6	47.38695° N 10.34438° E	1036 m	River
Allgäu Alps: Schochen	Trap 7	47.39202° N 10.36991° E	1930 m	Alpine grassland
Allgäu Alps: Schochen	Trap 8	47.39232° N 10.37057° E	1908 m	Spring
Allgäu Alps: Schochen	Trap 9	47.39368° N 10.36926° E	2032 m	South-exposed ridge with Blaugras-Horstseggenrasen
Allgäu Alps: Schochen	Trap 10	47.39307° N 10.36229° E	2010 m	South-exposed rock
Allgäu Alps: Schochen	Trap 11	47.39360° N 10.36615° E	1980 m	Snow bed
Allgäu Alps: Koblat	Trap 12	47.42223° N 10.34783° E	2160 m	South-exposed rock face
Allgäu Alps: Koblat	Trap 13	47.42147° N 10.35465° E	2033 m	Snow bed
Allgäu Alps: Koblat	Trap 14	47.42272° N 10.35730° E	2005 m	Mountain pine bush

**Table 2 insects-13-00082-t002:** Species numbers for 15 families of Diptera. Species numbers for the Bavarian and German faunas are shown for 11 families of Diptera with well-established taxonomy and for four families with limited knowledge (Cecidomyiidae, Chironomidae, Phoridae, Sciaridae). *—estimated at 80% of German fauna.

Taxon	Bavarian Species Count	German Species Count	Species Count Bavaria/Germany
Asilidae	68 [28]	85 [29]	0.80
Calliphoridae	50 *	62 [35]	0.80 *
Drosophilidae	64 [28]	81 [37]	0.79
Ephydridae	140 *	174 [38]	0.80 *
Muscidae	267 *	334 [48]	0.80 *
Sarcophagidae	107 *	134 [35]	0.80 *
Stratiomyidae	59 [28]	71 [30,48]	0.83
Syrphidae	389 [28]	458 [31]	0.85
Tabanidae	47 [28]	58 [8,48]	0.81
Tachinidae	361 [28]	501 [48]	0.72
Tipulidae	120 [33]	142 [32]	0.85
Cecidomyiidae	328 [38]	859 [5,6,7,8]	0.38
Chironomidae	576 [28]	781 [5,6,7,8]	0.74
Phoridae	302 *	378 [5,6,7,8]	0.80 *
Sciaridae	231 [28]	343 [43]	0.67
All Diptera	7635 *	9544 [8]	0.80 *

**Table 3 insects-13-00082-t003:** Sequence results for the three sampling sites. Total sample size, number of processed specimens, sequences recovered, BINs, BINs new to BOLD, Diptera specimens, and Diptera BINs.

	BFNP	ZSM	Allgäu Alps	Total
Samples (trap × collection events)	1 × 9 = 9	1 × 10 = 10	8 × 7 + 4 × 10 = 96	100
**All**				
Specimens	29,481	32,592	20,250	82,323
COI sequences (% success)	25,217 (85.6%)	28,923 (88.7%)	16,152 (79.8%)	70,293 (85.4%)
BINs (% new to BOLD)	2565 (19.4%)	3870 (15.8%)	4043 (23.0%)	8790 (23.8%)
**Diptera**				
Specimens (% of all specimens)	23,114 (78%)	15,448 (47%)	14,238 (70%)	52,800 (64%)
COI sequences (% success)	20,909 (91%)	14,983 (97%)	12,338 (87%)	48,230 (91%)
BINs (in % of all BINs)	1571 (61%)	1676 (43%)	2632 (65%)	4863 (55%)
Diptera BINs new to BOLD	375	260	736	1413
DT BINs new to BOLD (% of all new Diptera BINs)	337 (90%)	215 (83%)	215 (29%)	780 (55%)

**Table 4 insects-13-00082-t004:** Fifteen families of Diptera, 11 with well-developed taxonomy and four that are less well known. The number of BINs recovered in this study is followed by the known species count for Bavaria and Germany, the ratio of species counts for Bavaria and Germany, and BIN/Species ratios for Bavaria and Germany.

Taxa	BINs	Bavarian Species	German Species	Bavarian/German Species	BINs/Bavarian Species	BINs/German Species
Asilidae	13	68	85	0.80	0.19	0.15
Calliphoridae	22	50	62	0.80	0.44	0.35
Drosophilidae	27	64	81	0.79	0.42	0.34
Ephydridae	32	140	174	0.80	0.23	0.18
Muscidae	160	267	334	0.80	0.60	0.48
Sarcophagidae	35	107	134	0.80	0.33	0.26
Stratiomyidae	14	59	71	0.83	0.24	0.20
Syrphidae	131	389	458	0.85	0.34	0.29
Tabanidae	9	47	58	0.81	0.19	0.16
Tachinidae	126	361	501	0.72	0.35	0.25
Tipulidae	43	120	142	0.85	0.36	0.30
Average values					0.33 ± 0.9	0.27 ± 0.7
Cecidomyiidae	1163	328	859	0.38	3.55	1.35
Chironomidae	296	576	781	0.74	0.51	0.38
Phoridae	348	302	378	0.80	1.15	0.92
Sciaridae	339	231	343	0.72	1.47	0.99
Average values					1.67 ± 0.9	0.91 ± 0.3
All Diptera	4863	7635	9544	0.80	0.64	0.51

**Table 5 insects-13-00082-t005:** Proportion of undocumented Diptera biodiversity for Bavaria and Germany based on Chao1 estimates for 15 families.

Taxon	BINs	Chao1	BIN/Chao1	Bavarian Species	German Species	Chao1/ Bavarian Species	Chao1/ German Species
Asilidae	13	16	0.81	68	85	0.24	0.16
Calliphoridae	22	28	0.79	50	62	0.56	0.45
Drosophilidae	27	38	0.71	64	81	0.59	0.47
Ephydridae	32	88	0.36	140	174	0.63	0.51
Muscidae	160	220	0.73	267	334	0.82	0.66
Sarcophagidae	35	41	0.85	107	134	0.38	0.31
Stratiomyidae	14	16	0.88	59	71	0.27	0.23
Syrphidae	131	158	0.83	389	458	0.41	0.34
Tabanidae	9	10	0.90	47	58	0.21	0.17
Tachinidae	126	153	0.82	361	501	0.42	0.31
Tipulidae	43	59	0.73	120	142	0.49	0.42
Average values						0.46 ± 0.2	0.37 ± 0.2
Cecidomyiidae	1163	1937	0.60	328	859	5.91	2.25
Chironomidae	296	479	0.62	576	781	0.83	0.61
Phoridae	348	432	0.81	302	378	1.43	1.14
Sciaridae	339	468	0.72	231	343	2.03	1.36
Average values						2.55 ± 1.7	1.34 ± 0.5
All Diptera	4863	6927	0.70	7635	9544	0.91	0.73

**Table 6 insects-13-00082-t006:** BINs and calculated estimates. Total number of BINs recovered for each family from all traps, our calculated estimates, number of recorded species, and potential amplitude of new records for Bavaria and Germany.

Dark Taxa	BINs	Estimates Bavaria	Bavarian Species	New Records Bavaria	Estimates Germany	German Species	New Records Germany
BIN/species ratio							
Cecidomyiidae	1163	1817–3524	328	1489–3196	2280–4307	859	1421–3448
Chironomidae	296	463–897	576	0–321	580–1096	781	0–315
Phoridae	348	544–1055	302	242–753	682–1289	378	304–911
Sciaridae	339	530–1027	231	299–796	665–1256	343	322–913
Chao1/species ratio							
Cecidomyiidae	1937	2129–4211	328	1801–3883	2653–5235	859	1794–4376
Chironomidae	479	526–1041	576	0–465	656–1295	781	0–514
Phoridae	432	475–939	302	173–637	592–1168	378	214–790
Sciaridae	468	514–1017	231	283–786	641–1265	343	298–922

## Data Availability

The datasets containing all sequence data are publicly available in three datasets on the Barcode of Life Data System: DS-BFNP, DS-ZSMTRAP and DS-ALGALPS.

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
