# Peer review of "Peering into the Darkness: DNA Barcoding Reveals Surprisingly High Diversity of Unknown Species of Diptera (Insecta) in Germany"

_insects, 2022, doi:10.3390/insects13010082_

Round 1

Reviewer 1 Report

About the Introduction: the paper is about taxonomy—undetected species in southern Germany, compared to the known diversity. The known diversity, which generated a huge part of the data used along the paper, did not come out of the blue. There is a long history of contributions in German science to produce the % of known species in the fauna. When one reads the Introduction, it is like the literature never existed. The numbers come out from the blue.

Also, while discussing the issue of the taxonomy impediment, the reasoning is weird: the impediment (which is not clearly defined anywhere in the paper) is said to be a barrier to DNA barcoding applications. The causes or factors for the constraint, however, is said to be the limited involvement of experts with DNA-based identifications. The underlying logic is flaw and the reasoning and wording must be rebuilt.

The comment on the limited involvement of taxonomic experts with DNA-based identification is at least unpolite. It sounds like the taxonomists are lazy. DNA-based identification is a new branch in taxonomy and it is spreading at a good pace as goals, methods, and results are becoming available and as prices go down. Not a good politics to attract new partners.

On the expression "dark taxa", its original use was by Page, but Hausmann et al.'s (2020) use of the term is more clear, refering to species-rich groups of small insects with most their diversity undescribed. 
Figure 1 is badly conceived. There is no reason to exclude any identification below family, whatever it may be. It can be a BIN that does not match previous species, but it most certainly belong to recognizable genera.

In Methods ("Data analysis"), it is said that "each sequence ≥ 300 base pairs (bp) was automatically assigned to a Barcode Index Number (BIN) if it showed close sequence similarity to a BIN already in BOLD." What does "close sequence similarity means"??? It makes no sense, a paper with a lot of calculation and an approach to numbers, to have loose definition like this.

I understand that there may be the need to cherry-pick families for comparisons and that 11 of them were chosen. But what are the formal criteria to pick these families? In other words, what does "well-established" means? I am sure that there are other fly families that are even more "well-established taxonomic systems". They are probably not large enough or well-known etc. A better definition of "well-known" must be provided. Or just say that they were picked more or less randomly to be used as references.

On line 165: If the evidence available points to 80-85%, there is no justification whatsoever to use 0.75. All table should be corrected using numbers supported by evidence.

The graphs with the Hill numbers are nice. But nobody knows what this Hill numbers issue is about. It is not discussed at all along the text.

My major concern is about the Discussion section. Part of the conclusion is questionable ("little is known") and overall it is superficial. Chironomid numbers in the overall paper differ quite substantially from the number of the other three families. This is not even addressed. Germany has a long and solid tradition working with Chironomidae taxonomy, especially the influence of Bavaria-based Ernst Fittkau and Friedrich Reiss, for example. This also applies to the contributions from Werner Mohrig, Frank Menzel and Kai Heller to the taxonomy of Sciaridae. It is like the number of known species in Germany came out of the blue. The historical context of the development of German dipterology and its impact on the proportion of new BINs is entirely ignored. This left the discussion restricted to numbers without any concern with explanation. There is enough information on the progress of dipterology in Germany to bring it to the discussion. 

Also, the Malaise trap bias is not addressed. I perfectly understand that there are some technical constraints and a decision was made to concentrate the sampling to Malaise traps. That is ok. But there should be a word to address how this impact a comparison to the known fauna size coming from all different sources of sampling techniques.

Author Response

First, we would like to thank the reviewer for taking his/her time in reviewing our manuscript. We have addressed (and resolved) all concerns that were raised as best as possible.

About the Introduction: the paper is about taxonomy—undetected species in southern Germany, compared to the known diversity. The known diversity, which generated a huge part of the data used along the paper, did not come out of the blue. There is a long history of contributions in German science to produce the % of known species in the fauna. When one reads the Introduction, it is like the literature never existed. The numbers come out from the blue.

We thank the reviewer for pointing this out. We have revised large parts of the Introduction to prevent false interpretations (see 49: However, both orders contain several species-rich families which have received less attention than others in Germany’s long history of taxonomic research ).

We have added information regarding the source of our species numbers (see 88: An important backbone to our calculations are species numbers inferred from essential contributions of Germany’s over 200-year-long history of taxonomy [see 5–8, 28–38])

Also, while discussing the issue of the taxonomy impediment, the reasoning is weird: the impediment (which is not clearly defined anywhere in the paper) is said to be a barrier to DNA barcoding applications. The causes or factors for the constraint, however, is said to be the limited involvement of experts with DNA-based identifications. The underlying logic is flaw and the reasoning and wording must be rebuilt.

The comment on the limited involvement of taxonomic experts with DNA-based identification is at least unpolite. It sounds like the taxonomists are lazy. DNA-based identification is a new branch in taxonomy and it is spreading at a good pace as goals, methods, and results are becoming available and as prices go down. Not a good politics to attract new partners.

We have changed our wording throughout the entire Introduction regarding taxonomists – by no means did we want to imply that they are “lazy” or the reason for nameless BINs. Also, we have provided a definition of the “taxonomic impediment”, and constructed a better link to its relationship in DNA barcoding applications (see 55:  This imbalance of few researchers but high species numbers still awaiting documentation is commonly referred to as the taxonomic impediment. Against the backdrop of a worldwide decline in insect abundance, the taxonomic impediment is an alarming constraint to biodiversity surveys [15–19]. One such constraint is noticeable in the framework of DNA barcoding applications, where species proxies (Barcode Index Numbers, BINs) often lack a linkage to a known species [20].)

On the expression "dark taxa", its original use was by Page, but Hausmann et al.'s (2020) use of the term is more clear, refering to species-rich groups of small insects with most their diversity undescribed. 

We have provided more information on the term “dark taxa” (see 61: Page [20] coined the term “dark taxa” for these nameless BINs, and in 2020, Hausmann et al. [1] used it to address species-rich, taxonomically challenging groups of insect families whose diversity remains mostly undescribed.)

Figure 1 is badly conceived.

Unfortunately, the reviewer’s comment regarding Figure 1 does not mention why the cenception is badly conceived.

There is no reason to exclude any identification below family, whatever it may be. It can be a BIN that does not match previous species, but it most certainly belong to recognizable genera.

We did not exclude any identifications below family, and are confused as to which part of the manuscript this comment is referred to. Regarding Figure 1, we do not aim at providing species identifications, but to simply provide an examplary representative of the family.

In Methods ("Data analysis"), it is said that "each sequence ≥ 300 base pairs (bp) was automatically assigned to a Barcode Index Number (BIN) if it showed close sequence similarity to a BIN already in BOLD." What does "close sequence similarity means"??? It makes no sense, a paper with a lot of calculation and an approach to numbers, to have loose definition like this.

We have added information regarding this topic (see 142: Each sequence ≥ 300 base pairs (bp) was automatically assigned to a Barcode Index Number (BIN) already in BOLD if sequence similarity based on the (RESL-) BIN algorithm was fulfilled [43]).

This algorithm is well-explained in the cited literature, therefore we did not explain this any further.

I understand that there may be the need to cherry-pick families for comparisons and that 11 of them were chosen. But what are the formal criteria to pick these families? In other words, what does "well-established" means? I am sure that there are other fly families that are even more "well-established taxonomic systems". They are probably not large enough or well-known etc. A better definition of "well-known" must be provided. Or just say that they were picked more or less randomly to be used as references.

Yes, these families where chosen more or less randomly, and we have made according changes in the text (see 161: We selected, more or less randomly, 11 dipteran families whose taxonomy and fauna have been intensively studied up to date in order to assess the fractions of the Bavarian and German faunas represented in our samples (Table 2).)

On line 165: If the evidence available points to 80-85%, there is no justification whatsoever to use 0.75. All table should be corrected using numbers supported by evidence.

We have revised all calculations to 80%.

The graphs with the Hill numbers are nice. But nobody knows what this Hill numbers issue is about. It is not discussed at all along the text.

Explanations have been added in the text (see 152: We also generated continuous diversity profiles that illustrated variation in three standard metrics of biodiversity, which are quantified by Hill numbers (q): species richness (q=0), Shannon diversity (q=1), and Simpson diversity (q=2) [34]. Hill numbers are a mathematically consolidated group of diversity indices which include relative species abundances in order to quantify biodiversity.)

My major concern is about the Discussion section. Part of the conclusion is questionable ("little is known") and overall it is superficial.

We have changed our wording throughout the discussion (see 291: Although members of the order Diptera comprise almost a third of Germany’s insect fauna, the true diversity of the four highly diverse families [1] examined in this study is likely much higher than previously assumed.)

Chironomid numbers in the overall paper differ quite substantially from the number of the other three families. This is not even addressed. Germany has a long and solid tradition working with Chironomidae taxonomy, especially the influence of Bavaria-based Ernst Fittkau and Friedrich Reiss, for example. This also applies to the contributions from Werner Mohrig, Frank Menzel and Kai Heller to the taxonomy of Sciaridae. It is like the number of known species in Germany came out of the blue. The historical context of the development of German dipterology and its impact on the proportion of new BINs is entirely ignored. This left the discussion restricted to numbers without any concern with explanation. There is enough information on the progress of dipterology in Germany to bring it to the discussion. 

We have made changes to the discussion accordingly to the reviewer’s concerns (see 353: Although Chironomidae is a dark taxon, extensive research concerning the systematics, taxonomy and nomenclature of European and neotropical species has and is being conducted at the Bavarian State Collection of Zoology (ZSM) by the late Ernst Fittkau (former director of the ZSM) and his students including Martin Spies, the current editor of the Chironomid Home Page [62]. We therefore expect that the chironomid fauna of Bavaria and Germany is well documented and that, in contrast to the other DT-families, a much lower amplitude of new species will be discovered in the following years of GBOL III.).

Also, the Malaise trap bias is not addressed. I perfectly understand that there are some technical constraints and a decision was made to concentrate the sampling to Malaise traps. That is ok. But there should be a word to address how this impact a comparison to the known fauna size coming from all different sources of sampling techniques.

We have addressed this issue as follows (see 368: Our results did reveal 3-fold differences in recovery success among the well-known families, being lowest for Asilidae and Tabanidae and highest for the Muscidae. In our study, we used Malaise traps as a source of insect material, because they enable sampling of high numbers of flying insects, especially Diptera. However, a bias favoring the sampling of some taxa over others is always present, meaning that the community captured with such traps does not depict the true insect community of a sampled site. Furthermore, the set-up of a Malaise trap in terms of site choice, orientation and above-ground-level is another source of bias, and these factors strongly influence sampling results. To incorporate such variations, we used different approaches for extrapolating species numbers including Chao1 estimate calculations which consider the unsampled taxa present at the sampling sites.)

We have also rewritten the Conclusion:

In this study, we aimed at estimating the number of species in the Bavarian and German faunas for four families of Diptera that are prime examples of “dark taxa”. Our estimates were inferred from the analysis of sequence data, reproducible genetic patterns, rather than on speculations. The confidence intervals on these estimates are broad (Table 5), reflecting the various factors that influence any effort to gauge species diversity. Despite our limited geographic sampling effort, our results strongly suggest that a surprisingly high proportion of Germany’s biodiversity is yet to be discovered.

Reviewer 2 Report

This manuscript employs DNA sequencing of the “barcode” gene to examine the diversity of Diptera in Germany, with focus on four small bodied taxa likely to exhibit great unknown diversity. The paper clearly written, well composed, and well organized. It provides valuable insight into the insect biodiversity of Germany, particularly that portion of it that is completely unknown (the “dark taxa”). The study is of broad interest and should be published. I have mostly minor comments, but a couple significant ones (e.g. regarding the ratio used for high estimates of diversity).

Specific comments:

  1. 55 Constrains progress with what?
  2. 109 more specific information in the text concerning the start and end dates of trap sampling would be helpful – at least what months they started and stopped. (along with the more detailed information in the appendix). Also, I June 23 isn’t really “Spring”, so this statement is a little misleading.

Fig. 2 what is the dotted line on the map?

  1. 151 RCore is not the name of the software.
  2. 180-181. For item 2, What is the justification for using the lowest ratio here? Some taxa may be less likely to be collected in Malaise traps and thus might have a relatively low ratio (e.g., Asilidae), and it doesn’t make sense to use such a taxon as a baseline. Perhaps a mean or overall ratio of the “well known” taxa?
  3. 183- Was this ratio then used to estimate the minimum diversity of dark taxa? This is not clear.

Figure 3 – taxon names should be larger

  1. 210, Table 4, Figure 3. It is difficult to know how to interpret the ratios of BINS to known species – because we have no idea what the overlap in species is between them. High ratios do not necessarily mean high species richness and low ratios do not necessarily mean low species richness (except ratios >1, strongly suggest the existence of undescribed, unknown species).

It is also not entirely clear what the BIN/Chao1 ratios tell us (Fig. 3, table 5). Again, Malaise traps to not sample all taxa equally.

Figure 3 seems to duplicate information in the Tables. I don’t think it is necessary, or really helps understand or visualize the results. The authors should choose 1 (figure or table) and relegate the other to a supplement.

  1. 229 what is theta?
  2. 323 – I would argue the opposite. At least some small bodied taxa are likely to experience greater dispersal than larger bodied taxa (e.g., cecidomyiidae can disperse impressively far despite small size and limited adult life span).
  3. 348 the prior value (?)
  4. 349-350 A high BIN/species ratio does not necessarily indicate high diversity (see above comment)

The point of showing the diversity profiles appears to be that extrapolated richness of dark taxa is far (significantly) greater than the empirical estimates of richness and that this is not the case for most of the better known taxa. It’s not clear that plots are needed for this (as q > 0 are not really being examined), but I don’t have strong feelings about this.

Table 6 – see previous comment about the upper diversity estimates (The ratios they are based on are rather arbitrary)

Why were so relatively few of the new BINs from the Alps site (where half of the new Dipter BINs came from) dark taxa relative to the other sites (Table 3)?

It feels kind of odd to be talking about BINs and not species. This does get around the issue of whether BINs reflect species or not, but what are BINs? How do they relate to species? They are being compared with species to calculate ratios, so the assumption is that they are species – if these ratios are to mean anything. This being said, I appreciated the discussion (p. 14) by the authors of these issues and the relationships between BINs and species – and I agree with them, they are most likely underestimating species richness by using BINs. Some of these issues might be hinted at more in the Introduction.

Author Response

This manuscript employs DNA sequencing of the “barcode” gene to examine the diversity of Diptera in Germany, with focus on four small bodied taxa likely to exhibit great unknown diversity. The paper clearly written, well composed, and well organized. It provides valuable insight into the insect biodiversity of Germany, particularly that portion of it that is completely unknown (the “dark taxa”). The study is of broad interest and should be published. I have mostly minor comments, but a couple significant ones (e.g. regarding the ratio used for high estimates of diversity).

First, we would like to thank the reviewer for taking his/her time in reviewing our manuscript. We are happy to hear that (s)he finds our study interesting and have addressed (and resolved) all concerns that were raised as best as possible.

Specific comments:

  1. 55 Constrains progress with what?

This has been elaborated (see 59: Against the backdrop of a worldwide decline in insect abundance, the taxonomic impediment is an alarming constraint to biodiversity surveys.)

  1. 109 more specific information in the text concerning the start and end dates of trap sampling would be helpful – at least what months they started and stopped. (along with the more detailed information in the appendix). Also, I June 23 isn’t really “Spring”, so this statement is a little misleading.

We have added more information in the text and removed the statement with “spring” (see 102: In 2012, one trap was operated from May to September in the Bavarian Forest National Park (BFNP), a conifer-dominated montane forest. In 2014, 12 Malaise traps were placed along an altitudinal transect (1036-2160 m) in the Allgäu Alps, ranging from the Oytal to the Schochen and Nebelhorn Mountains. Traps in lower altitudes (Oytal) were deployed in May, whereas those in higher altitudes (Schochen and Koblat) in June. All traps in the Allgäu Alps were operated until October. Finally, in 2017, one trap was deployed at the Bavarian State Collection of Zoology (ZSM-SNSB) in Munich, which is situated in a residential neighborhood rich in backyard gardens. This trap was operated from April to December.)

Fig. 2 what is the dotted line on the map?

We have added this information to Figure 2. The dotted line is the Nebelhorn Cable Car on the Nebelhorn mountain.

  1. 151 RCore is not the name of the software.

We have corrected this (see 158: All calculations were done in R version 3.3.6)

  1. 180-181. For item 2, What is the justification for using the lowest ratio here? Some taxa may be less likely to be collected in Malaise traps and thus might have a relatively low ratio (e.g., Asilidae), and it doesn’t make sense to use such a taxon as a baseline. Perhaps a mean or overall ratio of the “well known” taxa?

This is a excellent point, and we thank the author for addressing this issue. We have corrected all calculations throughout the manuscript by applying the mean of the well known taxa as an upper limit to our estimates.

  1. 183- Was this ratio then used to estimate the minimum diversity of dark taxa? This is not clear.

Yes, this was the method used to estimate the minimum diversity (see 262:  We divided the number of observed BINs by the (BIN or Chao1)/species ratio for all Diptera to calculate minimum species numbers.).

(see 267: As 339 Sciaridae BINs were recovered, the minimum species estimate for Bavaria was 530 (339/0.64), while the upper estimate was 1,027 (339/0.33). Similarly, the number of species in Germany could be estimated as ranging from 665 (339/0.51) to 1,255 (339/0.27) species.)

Figure 3 – taxon names should be larger

As suggested, we move Figure 3 to the Appendix, therefore, we only minimally enlarged the text.

  1. 210, Table 4, Figure 3. It is difficult to know how to interpret the ratios of BINS to known species – because we have no idea what the overlap in species is between them. High ratios do not necessarily mean high species richness and low ratios do not necessarily mean low species richness (except ratios >1, strongly suggest the existence of undescribed, unknown species).

This is understandable, therefore, we have changed our wording in the discussion (see 340:  The much higher ratios for the DT-families (average ranging from 1.67–2.55 for Bavaria and 0.91–1.34 for Germany) strongly suggest the presence of undescribed, unknown species.)

It is also not entirely clear what the BIN/Chao1 ratios tell us (Fig. 3, table 5). Again, Malaise traps to not sample all taxa equally.

We have an explanation of this ratio in the methods section (see 149: We then calculated the ratio between the observed number of BINs in our samples to the estimate of species richness generated by Chao1 to ascertain the proportion of species at the sampling sites that have not been capture by our Malaise traps and that await analysis.)

Furthermore, we have added the bias of Malaise traps in the discussion (see 368: Our results did reveal 3-fold differences in recovery success among the well-known families, being lowest for Asilidae and Tabanidae and highest for the Muscidae. In our study, we used Malaise traps as a source of insect material, because they enable sampling of high numbers of flying insects, especially Diptera. However, a bias favoring the sampling of some taxa over others is always present, meaning that the community captured with such traps does not depict the true insect community of a sampled site. Furthermore, the set-up of a Malaise trap in terms of site choice, orientation and above-ground-level is another source of bias, and these factors strongly influence sampling results. To incorporate such variations, we used different approaches for extrapolating species numbers including Chao1 estimate calculations which consider the unsampled taxa present at the sampling sites.)

Figure 3 seems to duplicate information in the Tables. I don’t think it is necessary, or really helps understand or visualize the results. The authors should choose 1 (figure or table) and relegate the other to a supplement.

Figure 3 was moved to the Appendix.

  1. 229 what is theta?

We do not understand the reviewer’s comment.

  1. 323 – I would argue the opposite. At least some small bodied taxa are likely to experience greater dispersal than larger bodied taxa (e.g., cecidomyiidae can disperse impressively far despite small size and limited adult life span).

This sentence has been removed, as it can be subject to discussion.

  1. 348 the prior value (?)

This has been corrected (see 350: By contrast, our analysis of 7,148 specimens revealed 1,163 BINs, a count for Bavaria which is 3-fold higher than the number of recorded species.)

  1. 349-350 A high BIN/species ratio does not necessarily indicate high diversity (see above comment)

Yes, this is correct, therefore we have changed the wording to “suggest”.

The point of showing the diversity profiles appears to be that extrapolated richness of dark taxa is far (significantly) greater than the empirical estimates of richness and that this is not the case for most of the better known taxa. It’s not clear that plots are needed for this (as q > 0 are not really being examined), but I don’t have strong feelings about this.

Table 6 – see previous comment about the upper diversity estimates (The ratios they are based on are rather arbitrary)

We have redone all calculations and Table 6 has been revised to accommodate these changes.

Why were so relatively few of the new BINs from the Alps site (where half of the new Dipter BINs came from) dark taxa relative to the other sites (Table 3)?

We can not provide an answer to this question because it was not the focus of this study. Further research would be needed to shed light into this.

It feels kind of odd to be talking about BINs and not species. This does get around the issue of whether BINs reflect species or not, but what are BINs? How do they relate to species? They are being compared with species to calculate ratios, so the assumption is that they are species – if these ratios are to mean anything. This being said, I appreciated the discussion (p. 14) by the authors of these issues and the relationships between BINs and species – and I agree with them, they are most likely underestimating species richness by using BINs. Some of these issues might be hinted at more in the Introduction.

Reviewer 3 Report

This paper is a very high quality paper for Diptera Diversity work. I read several times and can not provide some suggestions.

I hope the author could complete more the leading work in the future.

Author Response

This paper is a very high quality paper for Diptera Diversity work. I read several times and can not provide some suggestions.

We thank the reviewer for taking his/her time to review our manuscript. We are very happy to receive such positive feedback!

I hope the author could complete more the leading work in the future.

Thank you.

Round 2

Reviewer 1 Report

Most concerns about the first version of the manuscript were solved.